# Remdesivir-bisPropionate, a better derivative of remdesivir against SARS-CoV-2: Comparison of *in vitro* and *in vivo* PK/PD Study as well as its therapeutic potential

Ashok Chakraborty[1]*, Anil Diwan[2], Vijetha Chiniga[1], Vinod Arora[1], Yogesh Thakur[1], Jayant Tatake[1], Rajesh Pandey[1], Preetam Holkar[1], Neelam Holkar[1]

**1** AllExcel, Inc, West Haven, Connecticut, United States of America, **2** Nanoviricide Inc, Shelton, Connecticut, United States of America

* ashok.chakraborty@allexcel.com

## Abstract

FDA approved remdesivir, which was though very effective against SARS-corona virus in cell culture system but in human its efficacy was below 10%, as reported. The main reasons are due to the poor stability of remdesivir in presence plasma. In order to increase the protective strength of remdesivir we took couple of approaches, one, to make an alternative but better derivative of remdesivir as remdesivir bis-propionate, and the other is to use our platform- designed biopolymer (NV387) to protect remdesivir compound from degradation in presence of plasma. Here we present our results as: (1) Remdesivir-bP is much more stable *in vivo* compared to remdesivir alone. (2) Remdesivir-bP when encapsulated within biopolymer, NV387, its stability is further enhanced. (3) The antiviral activity is also increased against NL-63 infection to rat model, compared to naked and/or encapsulated remdesivir. (4) The antiviral efficacy of the remdesivir pro-drug, therefore, can be mathematically drawn as follows: remdesivir-bP-encapsulated > remdesivir-encapsulated > remdesivir-bP > remdesivir.

## Introduction

COVID-19, a communicable pandemic respiratory infectious disease, is caused by a single-stranded RNA virus (SARS-CoV-2), which is similar to SARS-CoV-1. However, SARS-CoV-2 is more virulent than SARS-CoV-1, as the former causes cytokine release, septic shock, and blood clot formation that ultimately cause death in some individuals [1,2]. As a mechanism of action the virus uses five different proteins, one is the spike protein "S", which helps the virus to bind with the host cell receptor, and the others are "M" membrane protein, "E" envelope protein, "N" nucleocapsid protein, and certain other accessory proteins [3]. The M protein maintains the viral integrity [4], while the E protein plays a vital role in assembling the corona virus. The "N" protein maintains the nucleocapsid structure into a helical assembly. The accessory

**Data availability statement:** All relevant data are within the paper.

**Funding:** The author(s) received no specific funding for this work.

**Competing interests:** The authors have declared that no competing interests exist.

**Abbreviations:** ARDS, Acute respiratory distress syndrome; DMAP, 4-Dimethylaminopyridine; DMSO, Dimethyl sulfoxide; HPLC-ELSD, high-performance liquid chromatography- Evaporative light scattering detector; EUA, Emergency Use Authorization (EUA); PFU, Plaque forming unit; RDV, Remdesivir;RDV-bP, RDV-bispropionate; RdRp, RNA-dependent RNA polymerase; SARS-CoV-2, Single-stranded RNA-virus; USFDA, United States Food and Drug Administration; WHO, World Health Organization.

proteins are responsible for viral-host interactions and for replication. The viral replication involves an RNA-dependent RNA polymerase (RdRp), helicase, exonuclease N and some accessory proteins [5–10]. Fig 1 represents the schematic structure of SARS coronavirus.

- **RDV discovery and introduction as antiviral agent:**

Remdesivir (RDV), Favipiravir, and Ribavirin are typical drugs that interfere with the function of RdRp and therefore could be potential in blocking the virus replication [11]. Other drugs in this category may include fluorouracil and acyclovir, however, their clinical trials for testing their efficacy have not yet been conducted yet [12].

- **Remdesivir (GS-5734): Its Structure, Chemistry and Pharmacology:**

RDV is an analog of the nucleoside adenosine and designed as a prodrug [13]. Formula-wise it is a 2-ethylbutyl ($2S$)-2-[[[($2R,3S,4R,5R$)-5-(4-aminopyrrolo[2,1-f] [1,2,4]triazin-7-yl)-5-cyano-3,4-dihydroxy-oxolan-2-yl-methoxy-phenoxy-phosphoryl] amino] propanoate which has been developed by Gilead Sciences [14]. The cyano group in remdesivir (RDV) offers the strong nucleophilic C-C bonding, while the phosphoramido group provides enhanced lipophilicity to the molecule. The phenoxy group presents in RDV molecule aids in improved lipophilicity and cell permeability [9,10].

The structure of RDV have been modified into their monophosphate form, as well as an ester or phophoramidate forms, too. These modifications increase the cellular permeability of RDV, and it is bio-transformed to the active nucleoside or nucleoside monophosphate form [15]. The United States Food and Drug Administration (USFDA) approved an Emergency Use Authorization (EUA) to remdesivir [16], and later-on fully approved it on October 22, 2020.

- **Pharmacology of RDV**

*In vivo,* RDV (GS-5734) gets phosphorylated to its other form, remdesivir triphosphate (RDV-TP, GS-443902). The RDV-TP, in place of ATP, incorporates into the RdRp complex and inhibits its polymerase activity [17–19]. In clinical studies, the single-dose (3–225 mg) of RDV was found non-toxic with tolerable effects [20]. RDV is absorbed fast with an elimination half-life of about one hour [20,21]. The significant metabolites of RDV are GS-704277, GS-441524-MP, and GS-441524 [21–23].

- **Limitation of Effectiveness of RDV in Human with Regard to SARS Virus Treatment**

In brief, the randomized controlled trial with RDV in human did not show any clinical improvement, nor a reduction in mortality from SARS-CoV-2 infection [24,25], In fact, FDA approved remdesivir, which was initially found effective against SARS-CoV in cell culture systems but showed less than 10% efficacy in humans, as reported.

- Conventional cell culture protocols fail to account for the complex pharmacokinetics of RDV *in vivo* [26].

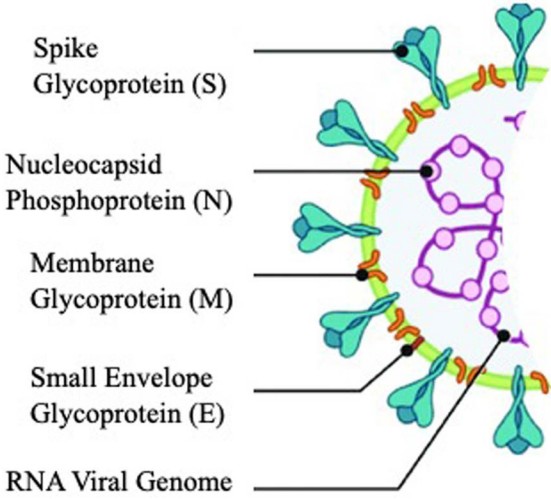

**Fig 1. Schematic Structure of SARS Coronavirus.**

- A non-uniform distribution of RDV, like high accumulation of GS-443902 and RDV metabolites in liver and kidney, were found after the drug administration in African green monkeys and cynomolgus macaques [27].

- Further, RDV is unstable in presence of plasma. In 5–30 min all the administered RDV were catabolized completely in presence of plasma *in vitro* [28], as well as *in vivo* [29,30]. The catabolism of RDV reduces the required exposure time of RDV to efficiently eliminate the virus.

- Further, on encapsulation of RDV within our platform-based discovery of biopolymer, NV-CoV-2 (its generic name NV-387), the antiviral activity of RDV remain sustained up to 24 hrs [30].

- Our pharmacokinetic studies with RDV in rat animals are similar with the reports made by Humeniuk et al. in humans [29].

- **Our approaches:**

Making a better RDV derivative with an increased stability in presence of plasma

Use of our Platform-technology based Biopolymer-NV-387 for Encapsulation of the Drug to increase the further stability of RDV-antiviral components in *in vivo* system for longer period of time.

## Materials and methods

### I: Preparation RDV-bispropionate (RDV-bP)

**Materials:** RDV, propionic anhydride, DMSO and 4-dimethylaminopyridine were procured fom Sigma-Aldrich, St. Louis. MO.

**Methods:** RDV-bP is synthesized by reacting 1 mole of RDV with 2.3 moles of propionic anhydride (stoichiometric excess) using 0.1 moles of DMAP as catalyst in DMSO. Reaction is carried out at room temperature until all the remdesivir molecule is converted to it's bis-ester. Reactions are monitored by MS. At the end of the reaction, RDV is completely consumed and mostly converted to the desired RDV-bP.

**Purification:** RDV-bP from the reaction mixture solution is precipitated with water. The resulting precipitate is washed repeatedly with water to remove residual DMSO and propionic acid. DMSO and propionic acid removal is monitored by

MS. Precipitate is dissolved in EtOH and dried at 50°C. Dried material is further characterized for Identification by MS and purity by HPLC.

## II: *In Vitro* Study: Stability assay of RDV and RDV-bP in Rat Plasma, *in vitro*

The samples, RDV in DMSO, RDV-bP, and RDV-SBECD (Gilead) were tested for their stability in presence of rat plasma *in vitro*. At indicated time points the reactions were stopped by adding acetonitrile extraction mixture and assayed for RDV and/or it's catabolite GS-441524 by LC-MS spectroscopy, as described previously in elsewhere [31].

## III: In Vivo study for Therapeutic potential of RDV-bP against SARS virus, toxicity and stability *in vivo*

- Animal Protocol: The entire animal handling experiments were done by Dr. Krishna Menon from AR Biosystems, (17633 Gunn Highway, Odessa, FL 33556), based on the protocol #IACUC No. 14/17ARB (Amended 17/19 & 3/20) for the study. This study followed strictly the guidelines and the recommendations from the "Guide for the Care and Use of Laboratory Animals of the National Institutes of Health." The number of animals used is considered the minimum necessary for meaningful biological comparisons and statistical calculations. The rats were observed daily for their body weight, clinical and behavior changes. The primary end points were survival and loss of body weight. All surgeries were performed under sodium pentobarbital anesthesia in order to minimize suffering. The rats were observed daily for body weight, clinical and behavior changes. At the end of the experiment, or when the animal comes at the moribund stages they were sacrificed by $CO_2$ asphyxiation. We are aware that all procedures involving experimental animals are ethically conducted and that they do not endure any unnecessary suffering.

For infection of animal with SARS virus we used NL-63 which infection-mechanism-wise similar to SARS-CoV-2 but less virulent, and permissible to use in BSL-2 laboratory like ours. HCoV-NL63 in which the amino-terminal part of the spike protein (S1) contains the receptor-binding domain, and the carboxy-terminal part (S2) contains a membrane spanning region. It was found that, NL63- and SARS-CoV-2 employ the same receptor, ACE2, for infectious cellular entry [32,33].

This was unexpected as NL63-S has no striking homology to either the whole S1 subunit of SARS-CoV or the already identified ACE2 interaction domain in SARS-CoV-S [34], suggesting that both proteins either form a common three-dimensional structure that allows ACE2 engagement in a similar fashion or that both S-proteins evolved different strategies to target ACE2.

Animal: ~200 g BW Rats, Sprague Dawley; Male and Female; 6–8 weeks (age); weight (160–190 grams) [Vendor, Taconic]. 6 animals per group were used.

Infection: hCoV-NL63, 20,000 PFU/120 uL for 200g animal, intratracheal

**Test Articles:** NV-387, NV-387-RP, RDV in DMSO, RDV in SBECD (Gilead), Placebo-Gummies, and Vehicle.

**Route of Administration (Table 1).**

- NV-387, NV387-RP and Placebo: Sublingual (P.O).

- Remdesivir and Vehicle for the same: I.V.

**Dosage Forms, Dosage amounts and frequency of dosing (Table 1).**

- **Experimental procedure**

Rats infected with human coronavirus, h-CoV-NL63, were used as a model of SARS CoV-2 infection. Like SARS CoV-2, h-CoV-NL63 binds to angiotensin-converting enzyme 2 (ACE2) receptor to enter into the cells. The animals were infected with approximately 20,000 pfu each of h-CoV-NL63 viral particles directly into lungs. All animals infected in this manner succumb to the disease in 6 days. The model also reproduces lung damage observed in SARS-CoV-2 infection in humans. Therefore, this model was selected as an animal model representative of COVID-19 disease in humans for the study.

**Table 1. Efficacy Testing of Test Articles in hCoV-NL63 Lung Infected Rats.**

| Test Articles | | | Dose Poly: RP (mg/animal) | No. Of Animals | Sex | Dose frequency/Day | Total Treatment Days |
|---|---|---|---|---|---|---|---|
| 1 | NV-387 | High | 55.5: 0 | 8 | Male | 1 per day, Days 0–9 | 10 |
| | | | | 8 | Female | | |
| 2 | | Medium | 27.75: 0 | 8 | Male | 1 per day, | 10 |
| | | | | 8 | Female | Days 0–9 | |
| 3 | | Low | 13.88: 0 | 8 | Male | 1 per day, | 10 |
| | | | | 8 | Female | Days 0–9 | |
| 4 | NV-387-RDV-bP | High | 31.1: 3 | 8 | Male | 1 per day, Days 0–9 | 10 |
| | | | | 8 | Female | | |
| 5 | | Medium | 15.55: 3 | 8 | Male | 1 per day, | 10 |
| | | | | 8 | Female | Days 0–9 | |
| 6 | | Low | 7.78: 3 | 8 | Male | 1 per day, | 10 |
| | | | | 8 | Female | Days 0–9 | |
| 7 | RDV-SBECD (Gilead) | | 3 x2 = 6 | 8 | Male | 2X Day 0, Once everyday following Days 1–9 | 11 |
| | | | | 8 | Female | | |
| 8 | Placebo for Gummies (Negative Control) | – | – | 6 | Male | 1 per day, Days 0–9 | 10 |
| | | | | 6 | Female | | |
| 9 | Vehicle for RDV (Negative Control) | | | 6 | Male | 2X Day 0, Once everyday following Days 1–9 | 11 |
| | | | | 6 | Female | | |
| 10 | Sham (Infected Untreated) | | | 4 | Male | 0 | 0 |
| | | | | 4 | Female | | |
| 11 | Untreated control | | | 2 | Male/Female | 0 | 0 |

- **Administration of test articles:**

**Injectables (*i.v*):** The Injectable Test Articles were administered on days and quantities as specified in **Table 1**. The administration was done by slow push injection in tail vein over a period of approximately 30 seconds.

   **Sublingual:** Test articles NV1108−397, NV1108–387-RDV-bP-and placebo for the gummy preparations were administered as a paste and painting the inner cavity of mouth by the use of a tuberculin syringe with no needle as per schedule and quantities in **Table 1**. Approximately half of the volume was fed at scheduled time. Then after a rest period of approximately ten minutes, the remaining volume was administered by the similar method.

## Results

**Identification:** Isolated product is identified by MS and MS(n). M.W. 714, m/z 715 (+ve ion) 713 (-ve ion).
   **Purity:** Purity by HPLC-ELSD is 99.2%.
   **Structure: (Fig 2)**
   **Characterization of RDV-bP: (Fig 3)**
   **Stability of RDV-bP in Presence of Rat Plasma (RPL) *in vitro* (Fig 4):** Extraction of RDV and it's metabolite GS, were done by using acetonitrile, as described in details elsewhere [31]. The results show that the RDV alone has a very

**Structure:**

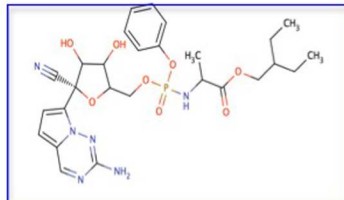

| Remdesivir | Remdesivir bispropionate |
|---|---|
| Mass = 602.5760 | Mass = 714.7025 |
| Exact mass = 602.225398638 | Exact mass = 714.277828138 |
| Formula = C27H35N6O8P | Formula = C33H43N6O10P |
| Isotope formula = C27H35N6O8P | Isotope formula = C33H43N6O10P |
| Atom count = 77 | Atom count = 93 |

| pH | logD |
|---|---|
| 1.50 | 1.87 |
| 5.00 | 2.09 |
| 6.50 | 2.09 |
| 7.40 | 2.09 |

| pH | logD |
|---|---|
| 1.50 | 4.15 |
| 5.00 | 4.37 |
| 6.50 | 4.37 |
| 7.40 | 4.37 |

**Fig 2. Structure of Remdesivir bispropionate.**

```
#:3   Ret.Time:Averaged 11.833-11.982(Scan#:1622-1643)
BG Mode:Averaged 9.753-14.161(1336-1946)
Mass Peaks:5   Base Peak:715.28(1257623)  MS Stage:MS  Polarity:Pos  Segment1 - Event1  Precursor:-----
```

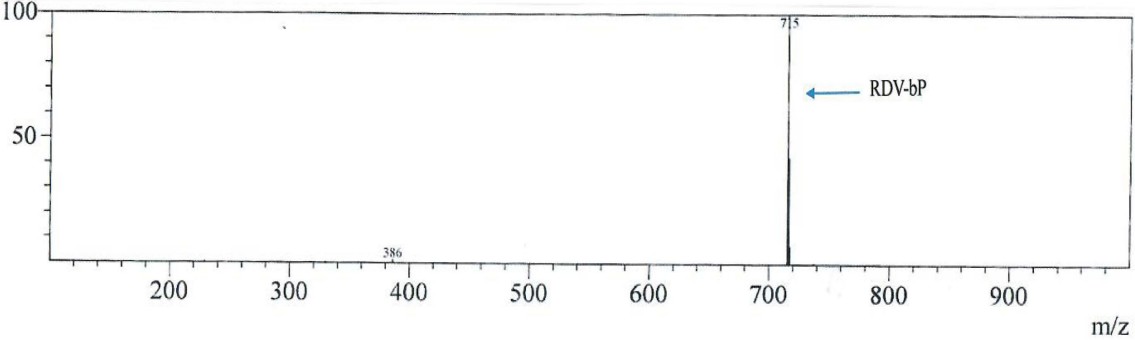

**Fig 3. Mass Spectroscopic Picture of RDV-bP.**

short life in presence of plasma, but RDV-bP has much longer half-life compared to the former. GS-441524 metabolite formations are representative of RDV breakdown, and supportive to each other data.

## Stability of RDV and RDV-bP with or without NV-387 encapsulation *in vivo*

We have compared the RDV levels in rat plasma by liquid chromatography-mass spectrometry (LC-MS) in a time-dependent manner after administration of the free RDV, NV387 polymer-encapsulated RDV (NV387-R), RDV-bP and polymer encapsulated RDV-bP (NV-387-RP) in rat as described in the following table. Gilead remdesivir was also used for comparison. RDV metabolite, GS-441524 was also measured to justify the RDV breakdown by rat plasma *in vivo* (**Fig 4**,**5**)

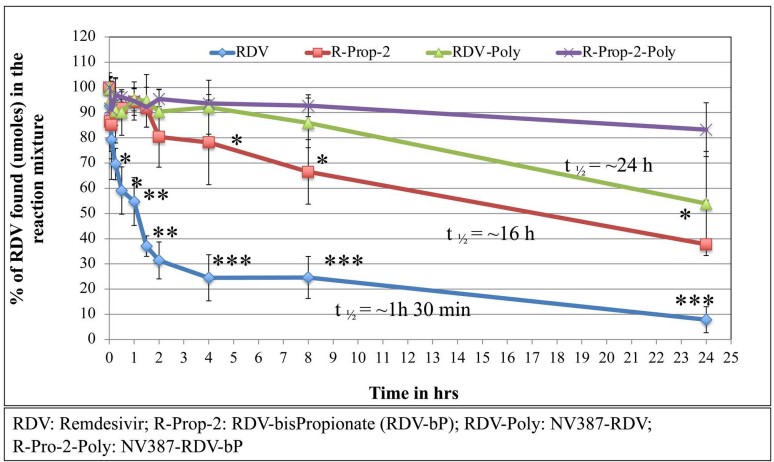

**Fig 4. Comparative stability analysis of RDV and RDV-bP with or without encapsulation in NV387 polymer and in presence of plasma, *in vitro*.** Statistical analyses were done from three independent experiments done in triplicate. *: p<0.5; **: p<0.01; p<0.001.

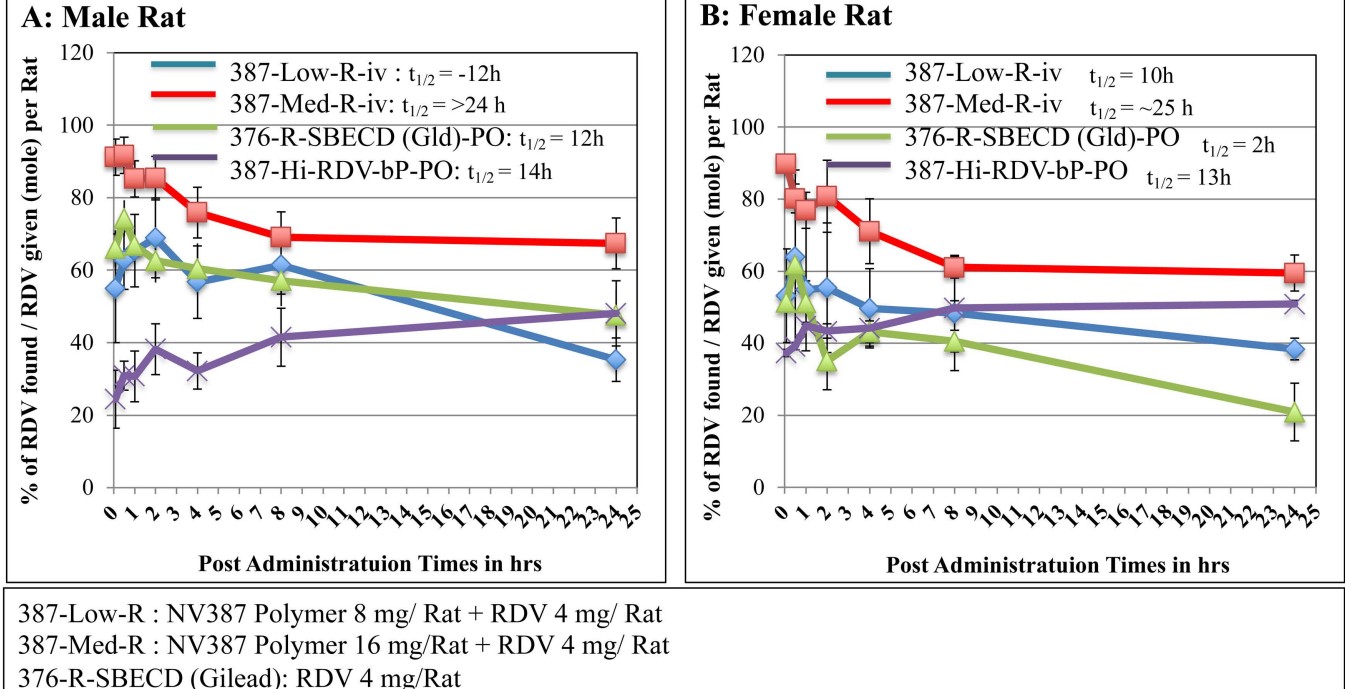

**Fig 5. Level of antiviral components (RDV) in Rat plasma (A: Male rat and B: Female rat) after 1st administration of different drug formulations with RDV.** Statistical analyses were done from two independent experiments done in duplicate. *: p<0.5; **: p<0.1; ***: p<0.05.

In fact, at up to 24 hours, the polymer-encapsulated RDV found stable, whereas free-RDV catabolized fully in 30 min *in vivo*. The Gilead RDV (RDV-in SBECD) showed the stability better than free-RDV ($t_{1/2}$ = 8 hrs) but did not last for long as RDV- NV387 ($t_{1/2}$ = 24 hrs). In brief, the stability grading is like this way: RDV-bP-encapsulated > RDV-encapsulated > RDV-bP > RDV. Through **Figs 4**,**5**, we have shown in both Male and Female Rat model animals, that polymer (NV387) encapsulated RDV is much more stable *in vivo* compared to naked RDV and/or Gilead RDV. The values of RDV-catabolite, GS441524, were also measured in all of the above experiments to show the proof of RDV stability or breakdown *in vivo (not shown here).*

## Therapeutic potential of RDV-bP against SARS virus, toxicity and stability *in vivo*

- **Observations**

Clinical observations were conducted twice daily for behavior, tone, ruffled hair, mucus membrane, behavioral changes, secretions, etc.

- **Body weight measurements**

Body weights were recorded for all animals, and randomized assignment of animals to dose groups was done using weight of animals, for the purpose of weight stratification. Prior to the first dose, body weights were recorded for all surviving animals continuing through the day of sacrifice. Only the day 0 body weights of animals were used for calculation of dosage for that animal. Necropsy body weights on the day of scheduled sacrifice were used for calculation of Organ to Body-Weight ratios to normalize for the variation in the specific organ weights.

- **Blood analysis**

One male and one female animal from each surviving group were bled to conduct blood analysis and clinical chemistry. There were no significant changes observed in any of the groups.

- **Histopathology**

One male and one female animal from each surviving group, and one that is moribund were sacrificed for gross histology and micro-histology of lungs, brain, heart, trachea, spleen, stomach, intestine, diaphragm, kidney and testes or uterus.

Organs were sliced using an American Optical Cryo Model 840 Cut 11 Microtome, H&E stained, and mounted for histopathology analysis.Slides were read out by the Study Monitor and the Study Director, and were further confirmed by the external lab of Dr. William Grey, DVM.

- **Antiviral effects of the Polymer (NV-387), polymer encapsulated-RDV and Polymer-encapsulated RDV-bP in rat animal model infected with NL-63 (Table 2; Fig 6)**

**Toxicity studies of RDV-bP:** Post mortem examination of the animals were conducted, taking one animal per group on day 4 and from the surviving animal from groups on day 10.

**Table 2. PK/PD/Antiviral Efficacy Evaluation of RDV_bP in comparison to NV-387 and NV387-RDV in Nl-63 infected Animals after daily dosing Sub-lingually using oral Gummies.**

| Description of the Agents | Polymer (NV 387) | Polymer (NV 387)-RDV-bP |
|---|---|---|
| | Actual wt (mg/200 g Rat) | Actual wt (mg/200 g Rat) |
| NV387- Low: RDV-bP | 15: 0 | 8: 3 |
| NV387- Low: RDV-bP | 30: 0 | 16: 3 |
| NV387- Low: RDV-bP | 60: 0 | 30: 3 |

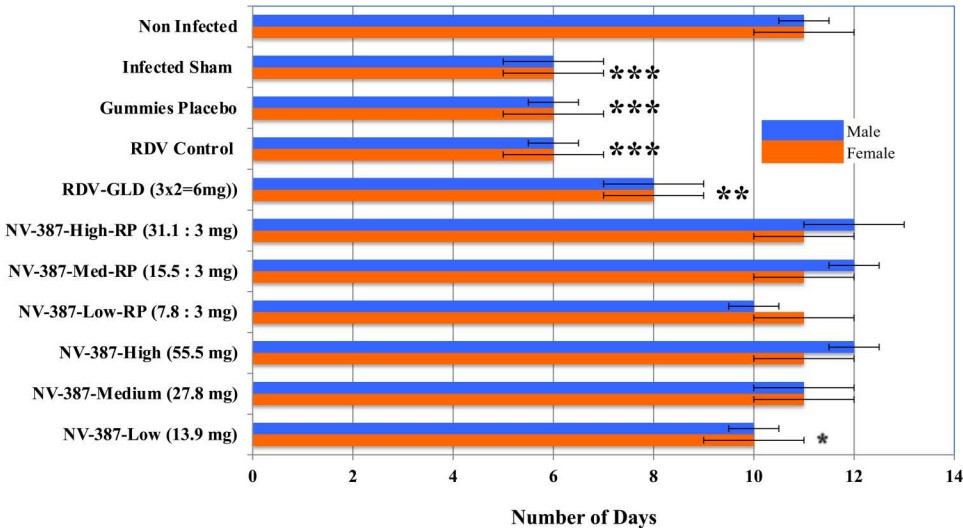

**Fig 6. Antiviral Efficacy of RDV-bP on NL63-Infected Rats: Survival of Male and Female rats infected with NL-63 SARS virus and treated with NV387, NV387-RDV, NV387-RDV-bP and RDV-SBECD (GLD).** Values are the mean±SD from 3 experiments. Statistical analyses were done from two independent experiments done in duplicate. *: $p < 0.5$; **: $p < 0.1$; ***: $p < 0.05$.

- **Results on day 4:** The animals in Sham Control showed signs of infection in the lungs. No lung exhhudate or secretion could be found. The weight of the lung is slightly increased, and there are signs of inflammation set in. Histological examination revels there is lung infection and inflammation. The lung alveoli is engorged, without any alveolar wall thickening. No visible signs of changes could be observed in any other organs. No other reportable changes observed in the H&E staining of brain, heart, trachea, spleen, stomach, intestine, diaphram, kidney, heart and testes or uterus.

The positive control Group (RDV-SBECD, Gld) had also infection started to set in. In the male animal lung, there was a slight reddish tinch at the ventrallobe of the lung. The weight of the lungs also inreased like that of the sham control. Histological examination revels there is lung infection and inflammation. The lung alveoli is engorged, without any alveolar wall thickening. No visible signs of changes could be observed in any other organs. No other reportable changes were observed in the H&E staining of brain, heart, trachea, spleen, stomach, intestine, diaphram, kidney, heart and testes or uterus. All other groups, one common occurance was the feacal pellets were not prominent and have slight yellow coloration giving the signs of loose motion to set in. This is attributed to the formulation materials and not seen when the administration of the compounds stopped.

The vehicle control group also showed signs of infection in the lungs with no lung exhudate or secretion. The weight of the lung is slightly higher with the beginning of setting the inflammation. Histological examination revels there is lung infection and inflammation. The lung alveoli is engorged, without any alveolar wall thickening. No visible signs of changes could be observed in any other organs. No other reportable changes were observed in the H&E staining of brain, heart, trachea, spleen, stomach, intestine, diaphram, kidney, heart and testes or uterus.

- **Results on day 10:** Both the oral groups had no changes observed, except that there are signs of early infection sets in the lungs histologically. The alveoli are less engorged, compared to the other two control groups but the weight of the lungs are slightly increased. Histological examination revels there is lung infection and inflammation. The lung alveoli is slightly engorged, without any alveolar wall thickening. No visible signs of changes could be observed in any other organs. No other reportable changes were observed in the H&E staining of brain, heart, trachea, spleen, stomach, intestine, diaphram, kidney, heart and testes or uterus.

All the iv treated groups did not show any change, except the feacal matter in the intestine. Histological examination revels there is no lung infection and inflammation. No visible signs of changes could be observed in any other organs. No other reportable changes were observed in the H&E staining of brain, heart, trachea, spleen, stomach, intestine, diaphram, kidney, heart and testes or uterus. The changes in weight gives a very positive reflection of the tolerability of the tested NV-387 polymer containing compounds. The changes in weight in these animals are due to the initial manupulation of the animals at the beginning of the experiment. The animals were well protected as the pattern of the change in the reduction of weight attributes to the progression of the diseaase. This gives a good indication that the compounds are well tolerated, and is a good candidates to take it for further development. BW measurements show no such toxicities of the compunds used by itself in this experiments (**Fig 7**, **8**).

## Discussion

The development of RDV, a broad-spectrum antiviral agent, has been a breakthrough for COVID-19 treatment. It is the first approved treatment for COVID-19 that is clinically used in about 50 countries [20]. RDV is approved to be the prodrug of nucleotide monophosphate. Such a prodrug is termed as protide [32,33]. The nucleotide monophosphate is linked to phosphoramidate ester with a phenyl group. This modification renders the monophosphates highly lipophilic. RDV penetrates into the intra cellular compartment and by the action of the enzyme esterase, undergo molecular cleavage to form the carboxylate [21,34]. The carboxylate undergoes cyclization, then it loses the phenyl group and gets transformed into phosphoramidase cleaves the phosphorus-nitrogen bond to leave the carboxyl amino group so as to regenerate the monophosphate. It then undergoes di-phosphorylation to become the triphosphate. The triphosphate binds to RdRp to terminate chain elongation and ultimately inhibits the viral replication [24,35].

However, the ultimate antiviral effect of RDV with regards to COVID-treatment does not reach to our expectation level. Further, nausea is the most common side effect of RDV, and it can further cause allergic reactions during its infusion or after the infusion. Besides, RDV with a dose of 150 mg for 7–14 days, causes a reversible increase of some liver enzymes like, alanine aminotransferase and aspartate transaminase [25,34]. RDV can also increase the prothrombin time [20,26].

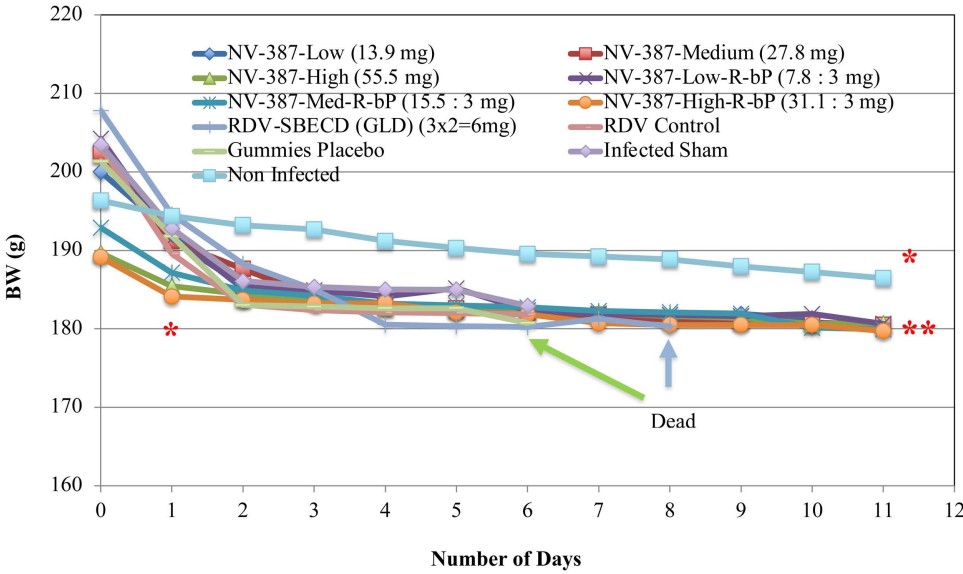

**Fig 7. Bodyweight profile of Male rats after the treatment with the drug components.** Statistical analyses were done from two independent experiments done in duplicate. *: p < 0.5; **: p < 0.05.

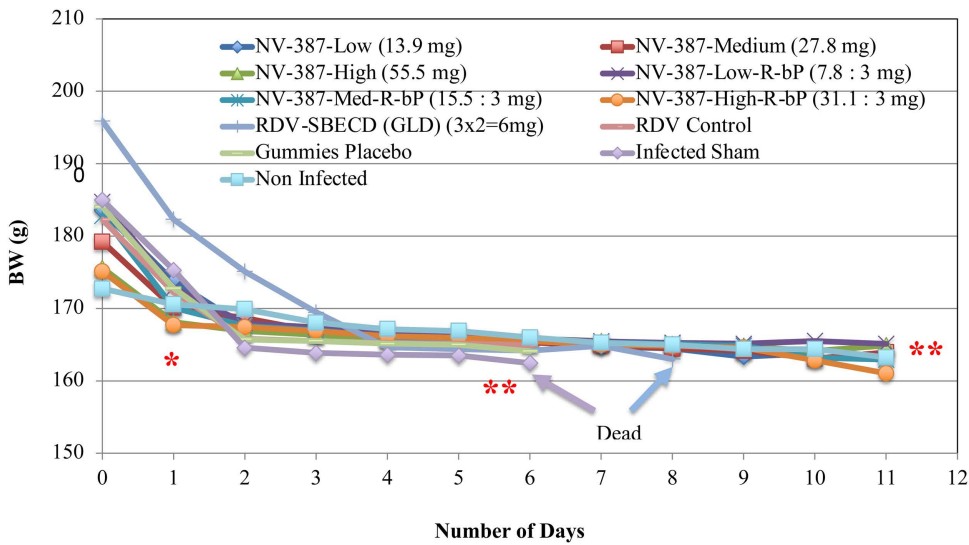

**Fig 8. Bodyweight profile of Female rats after the treatment with the drug components.** Statistical analyses were done from two independent experiments done in duplicate. *: p<0.5; **: p<0.05.

Further, the injection of RDV is prepared in sulfo-butyl-ether-b-cyclodextrin (SBECD), which can cause the renal dysfunction. However, no published report yet establishes any reproductive, teratogenic, developmental toxicity of RDV [20,32].

Therefore, a search for a better derivative of RDV is warranted, which can sustain in the *in vivo* system with or without encapsulation with the polymer for extended period or time. RDV-bispropionate is the result of our work, which shows improved stability *in vivo* with or without encapsulation into NV-387. Further, the antiviral efficacy in our experiment with rat model infected with NL63 were estimated as: NV387-RDV-bP>NV387-RDV>RDV-bP>RDV-SBECD (Gilead).

In histo-pathology of lungs, the lesions observed in all groups were consistent with viral infection: (1) Interstitial inflammatory cellular infiltration, (2) Thickened septa and inflammatory cellular infiltration, (3) Patchy areas of cellular consolidation, (4) Acute bronchiolitis surrounded by acute pneumonia. Post-mortem histological examination confirmed the same findings of lung infection, inflammation, and lung damage. The test APIs of this study, namely NV-387 and NV-387-RDV-bP are both broad-spectrum antivirals [36]. We have found NV-387 to be active against distinctly different coronaviruses, namely h-CoV-NL63 and h-CoV-229E (which uses APN as the cellular receptor) in cell culture studies. Thus, the efficacy of the test drug candidates NV-387 and NV-387-RDV-bP as observed in this model should be indicative of potential efficacy in human infections of SARS-CoV-2 and its variants.

Further, both the NV-387 and NV-387-RDV-bP were well tolerated in all treatment protocols studied. No serious adverse events were found. The changes in animals observed both physically/clinically and histologically represent changes due to viral infection. There was a slight increase in water content on the fecal examination, which is attributed to the test articles, that may be attributable to the water-retention properties of PEG present in the NV-387 itself.

In brief, RDV-bP has a better antiviral capacity *in vivo* than RDV itself and does not show any toxicity effects on the animal. The experiments were repeated several times with similar results. We have compared the RDV:GS ratio after 1st oral administration of encapsulated-RDV-bP (NV387-RDV-bP) in gummy form at 0 day and encapsulated-RDV by IV to Male and Female rats. In both the cases the amount of active RDV molecule administered was same in mg per rat. Note that, the ratio of RDV/GS is the determinant of the availability of the active antiviral component RDV in the system, and it's amount in blood is 500x more in case of RDV-bP compared to RDV alone even when encapsulated (Fig 6). It indicates that the stability and availability of active antiviral compound should be available more using RDV-bP instead of RDV itself.

Further, RDV-bP has an equal antiviral activity like RDV. Unlike NV376 (RDV-SBECD), NV387 and NV387-R at any doses despite of their higher accumulation in the body shows normal survival rate and body weight of the animal, indicating no toxicity of the drugs by itself, but effective as an antiviral agent.

These observations count the efficacy of RDV-bP as a better alternative and a suggestive pro-drug for SARS treatment. Besides, the use of NV-387 bio-nano-polymer should count a further improvement over Gilead RDV, in that regard. In addition, the inherent chemistry involved in this biopolymer (NV-387) will not let the virus to escape even it mutates.

A stable and effective method of drug delivery is also a subject for everybody's interest. Compared to marketed injectable formulation of RDV sublingual tablet of RDV having quality pharmaceutical properties has been invented [20,25]. Such tablets increase patient compliance in comparison to the former method of drug delivery. RDV-based masks [20] and cosmetics [29] have also been developed that help to prevent the spread of COVID-19. Accordingly, new spray [20,37] and nebulizer preparations [38] of RDV have been invented to address its side effects and improve its biological action. Some patents related to the combinations of RDV and other drugs, for example, cell pathway inhibitors (i.e., anti-inflammatory agents) [20], pyrimidine compound [39], isoxazoline parasiticide [40], azithromycin [41], and viral protease inhibitory polypeptide have been identified [42]. Such combinations have revealed the synergistic effects against COVID-19 compared to the monotherapy of RDV. Many combinations of RDV are in a clinical trial also [43]. The development of further inventive composition (monotherapy, combinations, and delivery systems) is anticipated to tackle this disaster of the century. It is now obvious from the present experiment that it offers a new delivery system along with a new active RDV analogs in terms of an improved bioavailability of the drug, it's biological effect, which can improve the patient compliance by reducing the side effects of the drug [44].

The compound patents of RDV are supposed to expire in October 2035. Gilead Sciences has also applied for the legal PTE (that may be up to five years) for its compound patents [20,45]. The extension of grant will extend the patent protection of RDV till October 2040 in many countries. The generic version of RDV may not be available to the patient by 2035–2040 in many countries. Accordingly, it is compulsory to look for alternative therapies also, e.g., drug repurposing of the existing therapeutic agents. Based on the above information on the future use of RDV as an antiviral drug, our new product RDV-bp and/or NV-387-encapsulated RDV-bP should be a promising one that to be considered. Computational screening of the spike-protein mutations in COVID patients may lead to design inhibitors against the virus binding [46]. In addition to that other ubiquitous inhibitors, like Protein kinase CK2 which enhances the viral protein synthesis should be explored [47]. Besides, anidulafungin and lopinavir are the most effective inhibitor of ACE2 receptor-S glycoprotein, therefore could be a potential drug for COVID-19 treatment [48].

## Conclusions

- The NNVC compounds (Nanoviricide™ company used actives) in this study can be broadly classified into four groups: the NNVC polymer (NV-387), RDV, Polymer encapsulated RDV (NV387-RDV), Polymer encapsulated RDV-bP (NV387-RDV-bP), and NV-377 (DMSO) is the vehicle controls. Besides, NV-376, the RDV formulation by Gilead (RDV-SBECD), as given in the clinics are used in these experiments. Sham controls received no treatment, except infusion with the virus.

- The "Sham controls" and the "Vehicle controls" had the life span of 5 days with the infection, where the RDV encapsulated form of the compound, when administered i.v, had life expansion more than tripled to 16 days, a remarkable achievement.

- The both concentrations of the NV387 also performed remarkably well, that it has increased the life span to 14 days.

- RDV alone achieved only a two and half days of life expansion.

- The oral administration of both NV-387 and NV387-RDV/ NV387-RDV-bP expanded the life span to 3 days and 4 days respectively.

- This is a good indication, that the compounds are effective orally, too.

- However, more studies are needed to bring out the optimal oral dosage for these compounds to achieve the same level of efficacy attained by the i.v. dosing.

## Acknowledgments

We acknowledge all our colleagues, Secretaries for their help during the preparation of the manuscript by providing all the relevant information.

## Author contributions

**Conceptualization:** Ashok Chakraborty, Anil Diwan, Vijetha Chiniga, Yogesh Thakur, Jayant Tatake.

**Data curation:** Ashok Chakraborty, Anil Diwan, Vijetha Chiniga, Yogesh Thakur, Jayant Tatake.

**Formal analysis:** Ashok Chakraborty, Vinod Arora, Rajesh Pandey, Preetam Holkar.

**Funding acquisition:** Anil Diwan.

**Investigation:** Ashok Chakraborty.

**Methodology:** Ashok Chakraborty, Vijetha Chiniga, Vinod Arora, Rajesh Pandey, Preetam Holkar, Neelam Holkar.

**Project administration:** Anil Diwan, Jayant Tatake, Preetam Holkar.

**Resources:** Anil Diwan, Vijetha Chiniga, Jayant Tatake, Rajesh Pandey, Preetam Holkar, Neelam Holkar.

**Software:** Preetam Holkar, Neelam Holkar.

**Supervision:** Anil Diwan, Jayant Tatake, Neelam Holkar.

**Validation:** Anil Diwan, Vijetha Chiniga, Vinod Arora, Yogesh Thakur, Jayant Tatake, Neelam Holkar.

**Visualization:** Anil Diwan, Vinod Arora, Yogesh Thakur, Jayant Tatake, Rajesh Pandey, Preetam Holkar, Neelam Holkar.

**Writing – original draft:** Ashok Chakraborty.

**Writing – review & editing:** Vinod Arora, Jayant Tatake.

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
