## [Decision Letter · Decision Letter 0]

2 Mar 2025

PONE-D-24-39848Remdesivir-bisPropionate, a better Derivative of Remdesivir against SARS-CoV-2: Comparison of in vitro and in vivo PK/PD Study as well as its Therapeutic PotentialPLOS ONE

Dear Dr. Chakraborty,

Thank you for submitting your manuscript to PLOS ONE. After careful consideration, we feel that it has merit but does not fully meet PLOS ONE’s publication criteria as it currently stands. Therefore, we invite you to submit a revised version of the manuscript that addresses the points raised during the review process.

We look forward to receiving your revised manuscript.

Kind regards,

Tamer M. Ibrahim

Academic Editor

PLOS ONE

Journal Requirements:

3. To comply with PLOS ONE submissions requirements, in your Methods section, please provide additional information regarding the experiments involving animals and ensure you have included details on (1) methods of sacrifice, (2) methods of anesthesia and/or analgesia, and (3) efforts to alleviate suffering.

Reviewers' comments:

Reviewer's Responses to Questions

**Comments to the Author**

1. Is the manuscript technically sound, and do the data support the conclusions?

Reviewer #1: Yes

Reviewer #2: No

Reviewer #3: Partly

2. Has the statistical analysis been performed appropriately and rigorously? 

Reviewer #1: No

Reviewer #2: N/A

Reviewer #3: No

3. Have the authors made all data underlying the findings in their manuscript fully available?

Reviewer #1: Yes

Reviewer #2: Yes

Reviewer #3: Yes

4. Is the manuscript presented in an intelligible fashion and written in standard English?

Reviewer #1: No

Reviewer #2: No

Reviewer #3: No

5. Review Comments to the Author

Reviewer #1: The manuscript entitled “Remdesivir-bisPropionate, a better Derivative of Remdesivir against SARS-CoV-2:Comparison of in vitro and in vivo PK/PD Study as well as its Therapeutic Potential” has many mistakes, authors need to rectify many portions.

o The introduction transitions abruptly between SARS-CoV-2 mechanisms, RDV pharmacology, and experimental approaches. Use connecting sentences or introductory phrases to guide the reader through the topics. Simplify or briefly summarize chemical details in the introduction, focusing on their relevance to the study.

o What specific data or references support the claim about RDV’s low efficacy in humans?

o Revise to: "FDA approved remdesivir, which was initially found effective against SARS-CoV in cell culture systems but showed less than 10% efficacy in humans, as reported."

o Variations in terminology, such as "RDV" vs. "Remdesivir," and inconsistent naming of derivatives like "Remdesivir-bP."

o Summarize ethical compliance in one concise paragraph.

o Add quantitative comparisons (e.g., half-life, stability metrics) alongside qualitative findings.

o Table-1 and Figure references are not accompanied by sufficient explanation in the text.

o The author required to update recent references can be seen PMID: 35014595, 36936534, 35401674, 33618621, 35265451

o Provide a short explanation of why NV387 was chosen and its anticipated benefits.

o Why does the introduction lack a proper background explaining why h-CoV-NL63 was chosen over SARS-CoV-2 itself as the model?

Present statistical significance (p-values, confidence intervals) for the stability grading claims.

o Specify which formulation components caused this side effect and how it could be mitigated.

Good Luck!

Reviewer #2: Review Report:

1. The abstract should be in a single paragraph with continuous form without breaking the sentences. Need to change the format.

2. The introduction part is not according to the journal style. So many sub headings in without objectives of the research work.

3. The materials and methods section, the authors just cut and paste of the student’s thesis, without following the PLOS ONE authors guideline.

4. The results and discussion part needs to rewritten.

5. Tables and figures are necessary. But the tables format needs to be changed as well as the clarity of the figures need to be improved.

Reviewer #3: Dear Author,

Thank you for submitting your abstract. Your study presents a promising approach to enhancing the stability and antiviral efficacy of remdesivir through the development of a bis-propionate derivative and the application of a protective biopolymer. Below are my comments and recommendations to improve clarity, scientific rigor, and adherence to journal requirements:

1. Title: The title effectively captures the focus of the study; however, refinement is needed to enhance clarity, conciseness, and scientific precision. Consider rewording to ensure it accurately reflects the study's scope and findings while maintaining readability.

2. Abstract: While the abstract presents the key aspects of the study, it does not fully align with the journal’s formatting and structural requirements. Ensure that it adheres to the prescribed format, including clear background, objective, methods, results, and conclusion sections, if required.

3. Introduction, Methods, Results, and Discussion: These sections require substantial revision to align with journal guidelines. The introduction should clearly define the research gap and rationale, the methods should provide sufficient experimental details for reproducibility, and the results and discussion should be presented in a logical and well-structured manner. Strengthening data interpretation and contextualizing findings within the existing literature will improve scientific rigor.

4. Overall Manuscript Quality: A more refined and structured revision is necessary to enhance readability, coherence, and alignment with journal standards. Consider revising the manuscript to improve clarity, logical flow, and adherence to scientific writing conventions.

I strongly recommend revising the manuscript in accordance with the journal’s formatting and content guidelines to improve its suitability for publication

6. PLOS authors have the option to publish the peer review history of their article (what does this mean? ). If published, this will include your full peer review and any attached files.

**Do you want your identity to be public for this peer review?** For information about this choice, including consent withdrawal, please see our Privacy Policy .

Reviewer #1: **Yes:** Shahzaib Ahamad

Reviewer #2: **Yes:** Sitesh Chandra Bachar

Reviewer #3: No

---

## [Author Response · Author response to Decision Letter 1]

14 Apr 2025

PONE-D-24-39848 [03/28/2025]

Remdesivir-bisPropionate, a better Derivative of Remdesivir against SARS-CoV-2: Comparison of in vitro and in vivo PK/PD Study as well as its Therapeutic Potential

PLOS ONE

Reviewer #1: The manuscript entitled “Remdesivir-bisPropionate, a better Derivative of Remdesivir against SARS-CoV-2: Comparison of in vitro and in vivo PK/PD Study as well as its Therapeutic Potential” has many mistakes, authors need to rectify many portions.

o The introduction transitions abruptly between SARS-CoV-2 mechanisms, RDV pharmacology, and experimental approaches. Use connecting sentences or introductory phrases to guide the reader through the topics. Simplify or briefly summarize chemical details in the introduction, focusing on their relevance to the study.

AC: Revised accordingly as shown below:

Introduction

COVID-19, a communicable pandemic respiratory infectious disease, is caused by a single-stranded RNA virus (SARS-CoV-2), which is similar to SARS-CoV-1. However, SARS-CoV-2 is more virulent than SARS-CoV-1, as the former causes cytokine release, septic shock, and blood clot formation that ultimately becomes cause fatal death in some individuals [1, 2]. As a mechanism of action the virus uses five different proteins, one is the spike protein “S”, which helps the virus to bind with the host cell receptor, and the others are “M” membrane protein, “E” envelope protein, “N” nucleocapsid protein, and certain other accessory proteins [3]. The M protein maintains the viral integrity [4], while the E protein plays a vital role in assembling the corona virus. The “N” protein maintains the nucleocapsid structure into a helical assembly. The accessory proteins are responsible for viral-host interactions and also for replication of virus [5, 6]. The viral replication involves an RNA-dependent RNA polymerase (RdRp), helicase, exonuclease N and some accessory proteins [5-8]. Fig.1 represents the schematic structure of SARS coronavirus.

o What specific data or references support the claim about RDV’s low efficacy in humans?

AC: The low EC50 values from in vitro studies did not reflect the drug’s clinical results, which was less than 10% efficacy in humans, as reported [33]. Further, the non-uniform distribution of remdesivir raises causes unwanted drug accumulation, potential toxicity, and the development of drug resistance discourage its use in human (Wang Z, Yang L, Song X-Q. Oral GS-441524 derivatives: next-generation inhibitors of SARS‐CoV‐2 RNA‐dependent RNA polymerase. Front Immunol. 2022;13:1015355. doi:10.3389/fimmu.2022.1015355).

o Variations in terminology, such as "RDV" vs. "Remdesivir," and inconsistent naming of derivatives like "Remdesivir-bP."

AC: Corrected

o Summarize ethical compliance in one concise paragraph.

AC: Done accordingly

o Add quantitative comparisons (e.g., half-life, stability metrics) alongside qualitative findings.

AC: Done accordingly

o Table-1 and Figure references are not accompanied by sufficient explanation in the text.

AC: Table-1 is mentioned in “Experimental procedure” section.

o The author required to update recent references can be seen PMID: 35014595, 35401674, 33618621, 35265451

AC: Now, Included in the discussion section with 3 new Refs.

o Provide a short explanation of why NV387 was chosen and its anticipated benefits.

AC: HCoV-NL63 was reported in 1981 and 1988 that the virus has been circulating and causing disease in the human population for a long time. This virus causes LRTIs and URTIs in 1.0–9.3% of children, the elderly and the immune-compromised, with symptoms ranging from mild to severe.

Current data clearly show that HCoV-NL63, which is BSL-2 compatible, share the same cellular receptor, ACE-2, similar to the most pathogenic BSL-2-incompatible SARS virus. Some similarities along with some dissimilarities of NL-63 and SARS-CoV in host cell entry mechanism and pathogenicity may open up a new strategy to find a proper therapeutics for the most virulent SARS virus.

o Why does the introduction lack a proper background explaining why h-CoV-NL63 was chosen over SARS-CoV-2 itself as the model?

AC: Now included in more appropriate position in the Materials and Method section where the infection methodologies were written.

o Present statistical significance (p-values, confidence intervals) for the stability grading claims.

AC: Done accordingly

o Specify which formulation components caused this side effect and how it could be mitigated.

AC: We didn’t notice and therefore didn’t mention any such side-effects.

Reviewer #2: Review Report:

1. The abstract should be in a single paragraph with continuous form without breaking the sentences. Need to change the format.

AC: Rewritten accordingly

2. The introduction part is not according to the journal style. So many sub headings in without objectives of the research work.

AC: Rearranged and rephrased.

3. The materials and methods section, the authors just cut and paste of the student’s thesis, without following the PLOS ONE authors guideline.

AC: Changed per advice of the reviewer.

4. The results and discussion part needs to rewritten.

AC: Changed per advice of the reviewer.

5. Tables and figures are necessary. But the tables format needs to be changed as well as the clarity of the figures need to be improved.

AC: Changed per advice of the reviewer.

Reviewer #3: Dear Author,

Thank you for submitting your abstract. Your study presents a promising approach to enhancing the stability and antiviral efficacy of remdesivir through the development of a bis-propionate derivative and the application of a protective biopolymer. Below are my comments and recommendations to improve clarity, scientific rigor, and adherence to journal requirements:

1. Title: The title effectively captures the focus of the study; however, refinement is needed to enhance clarity, conciseness, and scientific precision. Consider rewording to ensure it accurately reflects the study's scope and findings while maintaining readability.

AC: Title changes as: Remdesivir-bisPropionate, a better Derivative of Remdesivir to target SARS-CoV-2

2. Abstract: While the abstract presents the key aspects of the study, it does not fully align with the journal’s formatting and structural requirements. Ensure that it adheres to the prescribed format, including clear background, objective, methods, results, and conclusion sections, if required.

AC: Now it is rearranged as:

Abstract

FDA approved remdesivir, which was though very effective against SARS-corona virus in cell culture system but in human its efficacy was below 10%, as reported. The main reasons are due to the poor stability of remdesivir in presence plasma. In order to increase the protective strength of remdesivir we took couple of approaches, one, to make an alternative but better derivative of remdesivir as remdesivir bis-propionate, and the other is to use our platform- designed biopolymer (NV387) to protect remdesivir compound from degradation in presence of plasma. Here we present our results as: (1) Remdesivir-bP is much more stable in vivo compared to remdesivir alone. (2) Remdesivir-bP when encapsulated within biopolymer, NV387, its stability is further enhanced. (3) The antiviral activity is also increased against NL-63 infection to rat model, compared to naked and/or encapsulated remdesivir. (4) The antiviral efficacy of the remdesivir pro-drug, therefore, can be mathematically drawn as follows: remdesivir-bP-encapsulated > remdesivir-encapsulated > remdesivir-bP > remdesivir.

3. Introduction, Methods, Results, and Discussion: These sections require substantial revision to align with journal guidelines. The introduction should clearly define the research gap and rationale, the methods should provide sufficient experimental details for reproducibility, and the results and discussion should be presented in a logical and well-structured manner. Strengthening data interpretation and contextualizing findings within the existing literature will improve scientific rigor.

AC: Rearranged and rephrased.

4. Overall Manuscript Quality: A more refined and structured revision is necessary to enhance readability, coherence, and alignment with journal standards. Consider revising the manuscript to improve clarity, logical flow, and adherence to scientific writing conventions.

AC: Rearranged and rephrased.

I strongly recommend revising the manuscript in accordance with the journal’s formatting and content guidelines to improve its suitability for publication.

AC: Rearranged and rephrased.

---

## [Editor Report · Decision Letter 1]

2 May 2025

Remdesivir-bisPropionate, a better Derivative of Remdesivir against SARS-CoV-2: Comparison of in vitro and in vivo PK/PD Study as well as its Therapeutic Potential

PONE-D-24-39848R1

Dear Dr. Chakraborty,

We’re pleased to inform you that your manuscript has been judged scientifically suitable for publication and will be formally accepted for publication once it meets all outstanding technical requirements.

Kind regards,

Tamer M. Ibrahim

Academic Editor

PLOS ONE
---

## [Editor Report · Acceptance letter]

PONE-D-24-39848R1

PLOS ONE

Dear Dr. Chakraborty,

I'm pleased to inform you that your manuscript has been deemed suitable for publication in PLOS ONE. Congratulations! Your manuscript is now being handed over to our production team.

Kind regards,

on behalf of

Associate Professor Tamer M. Ibrahim

Academic Editor

PLOS ONE